# Analyzing the within and between Players Variability of Heart Rate and Locomotor Responses in Small-Sided Soccer Games Performed Repeatedly over a Week

**DOI:** 10.3390/healthcare10081412

**Published:** 2022-07-28

**Authors:** Ana Filipa Silva, Francisco Tomás González-Fernández, Rodrigo Aquino, Zeki Akyildiz, Luiz Palucci Vieira, Mehmet Yıldız, Sabri Birlik, Hadi Nobari, Gibson Praça, Filipe Manuel Clemente

**Affiliations:** 1Escola Superior Desporto e Lazer, Instituto Politécnico de Viana do Castelo, Rua Escola Industrial e Comercial de Nun’Álvares, 4900-347 Viana do Castelo, Portugal; anafilsilva@gmail.com; 2Research Center in Sports Performance, Recreation, Innovation and Technology (SPRINT), 4960-320 Melgaço, Portugal; 3The Research Centre in Sports Sciences, Health Sciences and Human Development (CIDESD), 5001-801 Vila Real, Portugal; 4Department of Physical Education and Sport, Faculty of Education and Sport Sciences, Campus of Melilla, University of Granada, 52006 Melilla, Spain; ftonzalez@ugr.es; 5LabSport, Department of Sports, Center of Physical Education and Sports, Federal University of Espírito Santo, Vitória 29075-910, Brazil; aquino.rlq@gmail.com; 6Sports Science Department, Gazi University, Ankara 06500, Turkey; zekiakyldz@hotmail.com; 7MOVI-LAB Human Movement Research Laboratory, School of Sciences, Graduate Program in Movement Sciences, Physical Education Department, UNESP São Paulo State University, Av. Eng. Luís Edmundo Carrijo Coube, 2085-Nucleo Res. Pres. Geisel, Bauru 17033-360, Brazil; luiz.palucci@unesp.br; 8School of Physical Education and Sports, Afyon Kocatepe University, Afyonkarahisar 03204, Turkey; mehmetyildiz@aku.edu.tr (M.Y.); sabribirlik38@gmail.com (S.B.); 9Department of Exercise Physiology, Faculty of Educational Sciences and Psychology, University of Mohaghegh Ardabili, Ardabil 5619911367, Iran; hadi.nobari1@gmail.com; 10Faculty of Sport Sciences, University of Extremadura, 10003 Cáceres, Spain; 11Department of Motor Performance, Faculty of Physical Education and Mountain Sports, Transilvania University of Braşov, 500068 Braşov, Romania; 12Sports Department, Universidade Federal de Minas Gerais, Belo Horizonte 31270-901, Brazil; gibson_moreira@yahoo.com.br; 13Instituto de Telecomunicações, Delegação da Covilhã, 1049-001 Lisboa, Portugal

**Keywords:** football, exercise monitoring, athletic performance, reproducibility

## Abstract

Background: Small-sided games (SSGs) are drill-based and constrained exercises designed to promote a technical/tactical and physiological/physical stimulus on players while preserving some dynamics of the real game. However, as a dynamic game, they can offer some variability making the prediction of the stimulus hardest for the coach. Aim: The purpose of this study was to analyze the between-session and within-player variability of heart rates and locomotor responses of young male soccer players in 3v3 and 5v5 small-sided game formats. Methods: This study followed a repeated-measures study design. Twenty soccer players were enrolled in a study design in which the SSG formats 3v3 and 5v5 were performed consecutively across four days. Twenty under-17 male youth soccer players (16.8 ± 0.4 years old) voluntarily participated in this study. Participants were monitored using a Polar Team Pro for measuring the heart rate mean and maximum, distances covered at different speed thresholds, and peak speed. Results: Between-players variability revealed that maximum heart rate was the outcome with a smaller coefficient of variation (3v3 format: 3.1% to 11.1%; 5v5 format: 6.6% to 15.2%), while the distance covered at Z5 (3v3 format: 82.5% to 289.8%; 5v5 format: 94.0% to 221.1%). The repeated measures ANOVA revealed that the four games tested were different in the within-player variability considering the maximum heart rate (*p* = 0.032), total distance (*p* < 0.001), and distances at zone 1, 2, and 5 of speed (*p* < 0.001). Conclusions: The smaller small-sided game tested promotes greater within-player variability in locomotor demands while promoting smaller within-player variability heart rate responses. Possibly, 5v5 is more recommended to stabilize the locomotor demands, while the 3v3 is recommended to stabilize the heart rate stimulus.

## 1. Introduction

Small-sided games (SSGs) are a widespread form of soccer practice given their capability to simulate both physical and technical–tactical demands at least in part similar to those required in official matches [1,2,3]. In the context of youth soccer, SSGs have been used for various purposes including training prescription, monitoring/testing, and possible talent detection [4,5,6]. However, aside from a number of additional strengths (e.g., exercise specificity and player buy-in), potentially undesired variability may be among the prominent pitfalls of SSGs, particularly when a training program aims to reach specific intensities (i.e., individualisation) and progress across the time. Otherwise, it may be an important advantage of such a tool since offering unpredictable stimuli generally benefits technical and tactical development [7,8]. As a consequence, it is clear that there is a need to know the level of variability promoted by distinct SSGs formats [9] before applying them in order to attempt to match the delivered stimuli with the coaches’ intentions.

Two common manners to check the consistency/inconsistency of SSG responses consist in determining their within- and between-session variability, and looking for capturing the possible day and seasonal oscillations through repeated measures, respectively. Importantly, the SSG variability data obtained in senior standards cannot be extrapolated to teenagers owing to the discrepancies previously detected [10,11]. This implies that the body of knowledge in this research area, notably in the former, i.e., more than half of existing studies (for a systematic review, see [8]), may not always apply to developing players. In addition to age, the time of day has some impact on young players’ performance based on the assumption that a single age group may encompass players with distinct chronotype profiles [12]. Although previous studies controlled such variables when testing within- or between-session variability of SSGs in youth soccer (e.g., performed testing at the same time across different days [13,14,15,16]), it remains questionable whether these may be representative for whole samples. As such, it is still necessary to gather information about SSG variability considering distinct periods of the day.

Research conducted on soccer players [15] indicates that either 2v2 or 4v4 SSGs are reproducible in under-19 (mean aged 16.3 years) players on a day-to-day and between-day basis whilst increases in game format tended to decrease control over the SSG intensity. In a follow-up study, contrasting results were reported for the same population and a similar experimental approach with no such evident influence of game format on the physiological variability of SSGs across time was used [9].

It is recognized that combining data collection and analysis of player effort, skill outputs, and psychological aspects (e.g., respectively, heart rate responses, ball handling, and enjoyment) is important in promoting a holistic approach to understanding SSGs [14,17,18,19]. While preliminary studies are available as aforementioned, with no consensus regarding SSG format effects on the variability of physiological parameters [9,15], the examination of additional features of SSGs such as technical–tactical and individual engagement is lacking in assessing its within- and between-session variability considering youth soccer athletes in particular [8]. It is particularly important to control the within- and between-player variability since the stimulus and volume of the demands imposed can propitiate asymmetries in the dose provided to each player which may expose players to under or over-stimulus. A consistent under- or over-exposure to training load may enhance the risk of bad adaptations or ultimately increase the risk of injury. Understanding within and between players’ variability in typical drills used in soccer training (as the SSGs) may help coaches to understand how to manage the implementation of SSGs and adjust them based on the players’ needs. Therefore, the present study aimed to investigate the within/between-session variability of physiological and locomotor responses during small-sided games (3v3 and 5v5) in young male soccer players.

## 2. Materials and Methods

### 2.1. Experimental Approach to the Problem

This study followed a repeated-measures study design. Twenty soccer players were enrolled in a study design in which the SSGs formats 3v3 and 5v5 were performed consecutively across four days (Table 1). The study was performed early in the season, two weeks after the beginning of the season. The study occurred after 24-h of rest (regarding the latest match), and was performed in the following environmental conditions: 3v3, day 1, (time period: 17:00 h; temperature:17 °C; relative humidity: 65%); 3v3, day 2, (time period: 17:00 h; temperature: 15 °C; relative humidity: 65%); 3v3, day 3, (time period: 17:00 h; temperature: 18 °C; relative humidity: 50%); 3v3, day 4, (time period: 17:00 h; temperature: 19 °C; relative humidity: 48%); 5v5, day 1, (time period: 17:00 h; temperature: 21 °C; relative humidity: 65%); 5v5, day 2, (time period: 17:00 h; temperature: 16 °C; relative humidity: 53%); 5v5, day 3, (time period: 17:00 h; temperature: 18 °C; relative humidity: 65%); 5v5, day 4, (time period: 17:00 h; temperature: 20 °C; relative humidity: 55%). 

### 2.2. Participants

Participants were enrolled based on convenience sampling. The sample size was calculated a priori in G*power (version 3.1.9.6.). For a power of 0.80 and an effect size of 0.5 (medium), the recommended total sample was 21. From the initial recruitment of twenty-four players, twenty male youth soccer players (16.8 ± 0.4 years old; 6.4 ± 0.7 years of experience; 167.9 ± 3.4 cm of stature; 65.4 ± 6.4 kg of body mass) belonging to the same team competing in the regional/national level voluntarily participated in this study. Four players from the initial twenty-four were excluded due to them being goalkeepers (n = 3) and missed one assessment (n = 1). The players were typically involved in five training sessions per week, plus one official match. Each training session lasted about 90 min. The eligibility criteria for being enrolled in the study were: (i) an outfield player; (ii) not injured in the last month before the experimental data collection and not present any signals or reports of injury or illness on the days of data collection; (iii) did not miss any of the data collection moments or repetitions performed; and (iv) did not take any drugs or energy drinks or changed any of their daily dietary routines.

### 2.3. Physical Fitness Assessment

The participants were preliminarily analyzed for their final locomotor profile. In the week before starting the observation, they were tested for the maximal sprint speed in a 30-m sprint test and for their ability to perform repeated efforts in the 30–15 Intermittent Fitness test. The data collection was preceded by a 24-h rest and occurred in the afternoon (5 pm). The players performed a standardized warm-up protocol (FIFA 11+). After that, they rested for 3 min and started the 30-m sprint speed assessment. They started with a split position (using the preferred leg in front), and the peak speed was assessed using a Polar Team Pro GPS (10 hz, Polar, Kempele, Finland) which was confirmed for the validity and reliability to estimate their peak speed in the previous study. After performing two trials of the 30-m sprint (interspaced by 5 min rest), they recovered for 3 min and followed the 30–15 Intermittent Fitness tests. The original protocol was implemented, and the final velocity completed was registered as the main outcome. For the case of maximal sprint speed, the best speed in the two trials was considered the main outcome.

### 2.4. Small-Sided Games

The formats 3v3 and 5v5 without a goalkeeper were implemented based on the characteristics presented in Table 1. Players were grouped into teams by their coaches. The same teams faced the same opponents at all times (bouts and different days) aiming to minimize the contextual variation influence. All the formats were preceded by the FIFA11+ warm-up protocol [20]. Four balls were positioned around the pitches aiming to quickly replace the ball in case of going out of boundaries. No verbal encouragement was conceded during the games. 

### 2.5. Heart Rate (HR) and Locomotor Demands

The heart rate and locomotor demands were monitored using the Polar Team Pro (Polar, Finland) which is confirmed for its reliability to measure the demands analyzed [21,22]. The sensor was positioned in the center of the chest while using a band. Each player always used the same unit in order to avoid inter-unit variability. The following measures were obtained per game: minimum heart rate (HRmin); average heart rate (HRmean); peak heart rate (HRpeak); peak speed; average speed; distance covered per minute; distance covered at zone 1 (Z1: 3.00 to 6.99 km/h) per minute; distance covered at zone 2 (Z2: 7.00 to 10.99 km/h) per minute; distance covered at zone 3 (Z3: 11.00 to 14.99 km/h) per minute; distance covered at zone 4 (Z4: 15.00 to 18.99 km/h) per minute; distance covered at zone 5 (Z5: >19.00 km/h) per minute.

### 2.6. Statistical Procedures

The descriptive statistics were presented in form of mean and standard deviation. The coefficient of variation (expressed as a percentage) was calculated to analyze within-player variability (variability of the player across different days for the same condition) and between-player variability (heterogeneity of the responses for a specific game and condition). The normality of the sample was not verified in all the measures; however, since the data were greater than 30, we have assumed normality by the Central Limit Theorem. A repeated measures ANOVA was performed to compare the within-players variability obtained in each game (format and pitch dimension). Moreover, for each format of play and pitch dimension, a repeated measures ANOVA was tested (aiming to analyze between training-days variation). Partial eta squared was used to calculate the effect size. Bonferroni was used as a post hoc test in the pairwise comparisons. The statistical procedures were executed in the SPSS (version 28.0.0.0, IBM, Chicago, IL, USA) for a *p* < 0.05.

## 3. Results

The physical fitness assessment revealed that players presented a maximal speed sprint of 28.7 ± 2.8 km/h and a final velocity in the 30–15 Intermittent Fitness test of 15.2 ± 1.3 km/h. Descriptive statistics of between-training sessions variability can be found in Table 2 for 3v3 and Table 3 for 5v5 formats. While the maximum heart rate presented in the smaller coefficient of variation in 3v3 played at 39 × 24 m (3.2% to 11.1%) and 32 × 19 m (3.1% to 8.9%), the distance covered at Z5 presented the greater coefficient of variation in 3v3 played at 39 × 24 m (82.5% to 289.8%) and 32 × 19 m (91.1% to 195.7%). 

Similarly, the maximum heart rate presented a smaller coefficient of variation in 5v5 played at 50 × 31 m (6.6% to 16.9%) and 40 × 25 m (7.0% to 15.2%) and a greater coefficient of variation in distance covered at Z5 in 5v5 played at 50 × 31 m (103.7% to 221.1%) and 40 × 25 m (94.0% to 156.8%).

Table 4 presents the descriptive statistics of the within-player variability across the different training sessions performed for each format of play. Additionally, Figure 1, Figure 2, Figure 3, Figure 4 and Figure 5 present the within-player variability in a graphical representation. The repeated measures ANOVA tested the differences in coefficient of variation between formats and pitch dimensions. The repeated measures ANOVA revealed that the four games tested were different in the within-player variability considering the maximum heart rate (*p* = 0.032; $\eta_{p}^{2}$ = 0.172), the total distance (*p* < 0.001; $\eta_{p}^{2}$ = 0.441), the distance at Z1 (*p* < 0.001; $\eta_{p}^{2}$ = 0.745), the distance at Z2 (*p* < 0.001; $\eta_{p}^{2}$ = 0.444), and the distance at Z5 (*p* < 0.001; $\eta_{p}^{2}$ = 0.394). The 3v3 played at 39 × 24 m presented the greater CV% in total distance (34.8 ± 10.5), distance at Z1 (60.3 ± 13.8), the distance at Z2 (52.8 ± 14.0), and the distance at Z5 (170.5 ± 27.6). On the other hand, the 5v5 played at 50 × 31m presented a smaller CV% in the total distance (18.4 ± 8.4), the distance at Z1 (19.6 ± 10.1), the distance at Z2 (32.2 ± 13.5), and the distance at Z5 (113.7 ± 38.5). 

The between-session analysis for each game was performed using a repeated measures ANOVA. In the case of format 3v3 played at 39 × 24 m (Table 5), significant differences were found between sessions on HRmean (*p* = 0.001; $\eta_{p}^{2}$ = 0.303), HRpeak (*p* = 0.023; $\eta_{p}^{2}$ = 0.189), total distance (*p* = 0.005; $\eta_{p}^{2}$ = 0.281), distances at Z1 (*p* = 0.001; $\eta_{p}^{2}$ = 0.422), Z2 (*p* = 0.007; $\eta_{p}^{2}$ = 0.255), Z4 (*p* = 0.006; $\eta_{p}^{2}$ = 0.196), Z5 (*p* = 0.002; $\eta_{p}^{2}$ = 0.324), and peak speed (*p* < 0.001; $\eta_{p}^{2}$ = 0.330). In the case of 3v3 played at 32 × 19 m (Table 5), significant differences were found between sessions on the total distance (*p* = 0.003; $\eta_{p}^{2}$ = 0.301), distances at Z1 (*p* = 0.007; $\eta_{p}^{2}$ = 0.259), Z2 (*p* = 0.001; $\eta_{p}^{2}$ = 0404), Z4 (*p* = 0.009; $\eta_{p}^{2}$ = 0.220), and peak speed (*p* = 0.037; $\eta_{p}^{2}$ = 0.180).

The between-session analysis for each game was performed using a repeated measures ANOVA. In the case of format 5v5 played at 50 × 31 m (Table 6), no significant differences between sessions were found for any of the main outcomes. In the case of 5v5 played at 40 × 25 m (Table 6), significant differences were found between sessions on the total distance (*p* < 0.001; $\eta_{p}^{2}$ = 0.291), distances at Z1 (*p* < 0.001; $\eta_{p}^{2}$ = 0.377), Z4 (*p* = 0.008; $\eta_{p}^{2}$ = 0.225), Z5 (*p* < 0.001; $\eta_{p}^{2}$ = 0.438), and peak speed (*p* = 0.012; $\eta_{p}^{2}$ = 0.174).

## 4. Discussion

The current research reveals that locomotor demands present greater between-session and within-player variability than heart rate responses. Additionally, it was found that 3v3 small-sided game formats promote smaller within-player variability in heart rate responses, yet promote greater locomotor demand variability than the 5v5 format. 

One of the concerns in training prescription is projecting the dose for the players. The act of designing small-sided games is complex since many task constraints can be used in combination and some of them can contribute to changing behaviors that are naturally associated with variation in players’ responses [6]. In our study, we tried to identify how physiological and locomotor players’ responses can vary across four consecutive days for the same games. The within-player variability revealed that heart rate responses are more stable, and the coefficients of variation are, on average, between 6.0 and 11.9% for maximum and mean heart rate. These values are slightly higher than those reported by previous studies on the variability of heart rate responses in the same small-sided games [4,15,23], yet they confirm that physiological responses are relatively reproducible across the days and conditions. The games played no effect on the within-players variability in heart rate mean, although significant differences were found between games on the maximum heart rate. On average, the 3v3 varied between 6.0% and 6.5% of the coefficient of variation, while 5v5 games ranged from 10.3% to 10.4%. The fact that smaller formats of play induce smaller variability can be associated with the time of repetition (3 min) and the fact that, in the 3v3 format, the levels of participation were more equal [24]. Interestingly, this finding may help coaches in the training prescription, since 3v3 induces values closer to those necessary for developing aerobic power [1], and these stable values presented by players across sessions can help coaches to trust in the 3v3 format to induce a similar physiological stimulus while prescribing small-sided games for targeting aerobic power [25].

On the other hand, locomotor demands presented great levels of within-players variability. While the total distance covered presented coefficients of variation between 18% and 35% in the games tested, the distances covered by different speed thresholds revealed a progressive increase of within-player variability. The distance covered at running intensity Z5 was about 113% and 171% in the games tested. These values and tendencies are aligned with previous studies [26,27] which suggested that distances covered with higher intensities are the same with greater within-player variability. Considering that the small-sided games shared the dynamics of the game and are dependent on the tactical behaviors and contextual factors [28], it is expected that intensities can vary based on the moment and tactical dimension of the game [29]. Interestingly, it was observed that the 5v5 format presented a significantly smaller coefficient of variations for within-player variability (e.g., total distance covered, distances covered at Z1, Z2, and Z5) which suggests that possibly bigger formats of play may offer a more stable pattern of locomotor profile. This can be associated with the playing position which will be more structured in formats with a greater number of players as in 5v5. In the case of 3v3, the players will not have a specific position (or, at least, not so structured) which will increase the participation from game to game. 

A descriptive observation allowed also us to identify the heterogeneity level in which the game (between players) was smaller in heart rate responses and greater in locomotor demands. Using small-sided games as the format of exercise for prescribing physical stimulus is challenging since the game is dynamic. The heterogeneity between players is expectable based on the dynamic of the game as well. Between-player variability varied from 3.2% to 11.1% in the worst case for the heart rate responses in the 3v3 games performed, while it varied from 7.0% to 19.7% in the 5v5 format. This means that 5v5 may induce a greater inter-player variability. This can be caused by the greater number of players participating in the game and by the fact that tactical positions assumed by the players may interact to explain these varieties in the physiological stimulus. However, greater variabilities occur in locomotor demands as in the case of distance covered at Z5 in 3v3 in which variabilities between players achieved 245% of the coefficient of variation, while in the 5v5, dropped to a maximum of 221.1%. Coaches must be attentive to this between-player variability since locomotor demands are commonly considered for the identification of neuromuscular stimuli. Considering that Z5 corresponds to the sprint zone, and keeping in mind that sprint is one of the most important neuromuscular stimuli for players, some players can suffer in small-sided games by under or over-stimulation.

### 4.1. Study Limitations

This study presents some limitations. The small sample included is one of the limitations that should be considered as a potential bias. Another limitation is convenience sampling. All the players were part of the same team, which should be faced as a bias for generalization of the evidence. Thus, any conclusion should not be observed as an irrefutable finding and requires more observations to be confirmed. Moreover, this study tested repeated measures on consecutive days which, in fact, can interfere with the way players respond day-by-day. However, this is also an issue for the practice in which specific formats are applied in different contextual scenarios. Another limitation is that some thresholds (heart rate, speed thresholds) are associated with physical fitness, which makes it harder to detect stable values. For example, speed thresholds are player-dependent which means that some thresholds (exempla as sprint) may not start at the same velocity for all. Future research should consider increasing the number of weeks observed (to have a real repeated measure considering the one-game format played on the same day of the week). Moreover, increases in the number of players will favor the generalization of the findings. Finally, other measures that can better describe the anaerobic contribution as blood lactate concentrations must be considered in future research to identify the influence of the most intense SSGs.

### 4.2. Practical Applications

The current research may provide some practical applications. In this case, the 3v3 format must be used by coaches for an aerobic power stimulus, since it seems to be more stable in terms of stress on heart rate. On the other hand, in the case of aiming to focus on locomotor demands, possibly adjusting the format for medium-sided games (as in 5v5) can be better, since the values are less variable across the sessions. 

## 5. Conclusions

The current research suggests that within- and between-player variability occurs in small-sided games; however, they are associated with the formats and task constraints implemented. The results indicate that heart rate responses are less variable in 3v3 formats, while locomotor demands are less variable in 5v5 formats. Additionally, it is observed that between-player variability is smaller in heart rate responses than in locomotor demands. The results come from a small sample and convenience sampling. Thus, these results are context-dependent which should be faced as a limitation for the generalization of the findings. Future studies should be conducted to confirm or refute the evidence. The next studies must include more players to increase the sample size. Additionally, different teams and populations must be included for a more robust analysis. Finally, consideration for moderators and mediators such as training status, skill level, and moment of the season must be also included.

## Figures and Tables

**Figure 1 healthcare-10-01412-f001:**
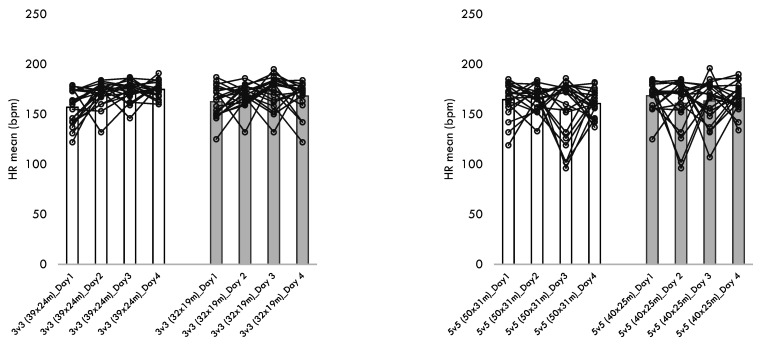
Descriptive statistics of heart rate (HR) responses over the different formats, pitch dimensions, and days.

**Figure 2 healthcare-10-01412-f002:**
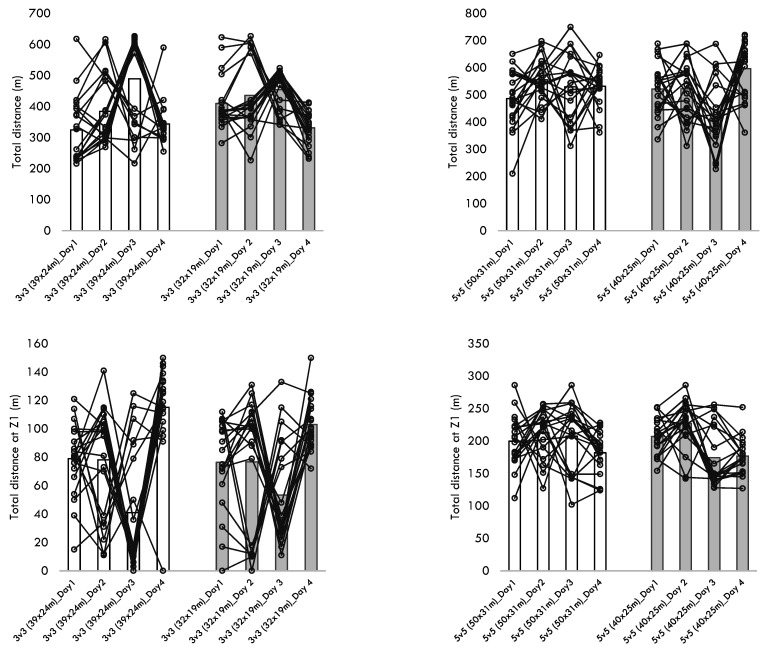
Descriptive statistics of total distance and distance covered at Z1 over the different formats, pitch dimensions, and days.

**Figure 3 healthcare-10-01412-f003:**
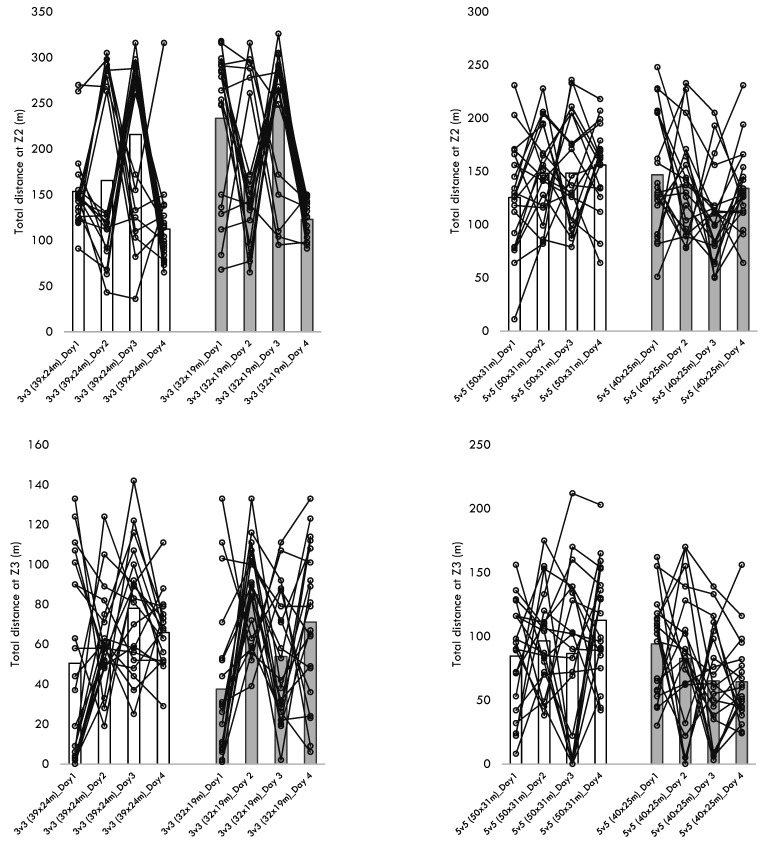
Descriptive statistics of distances covered at Z2 and Z3 over the different formats, pitch dimensions, and days.

**Figure 4 healthcare-10-01412-f004:**
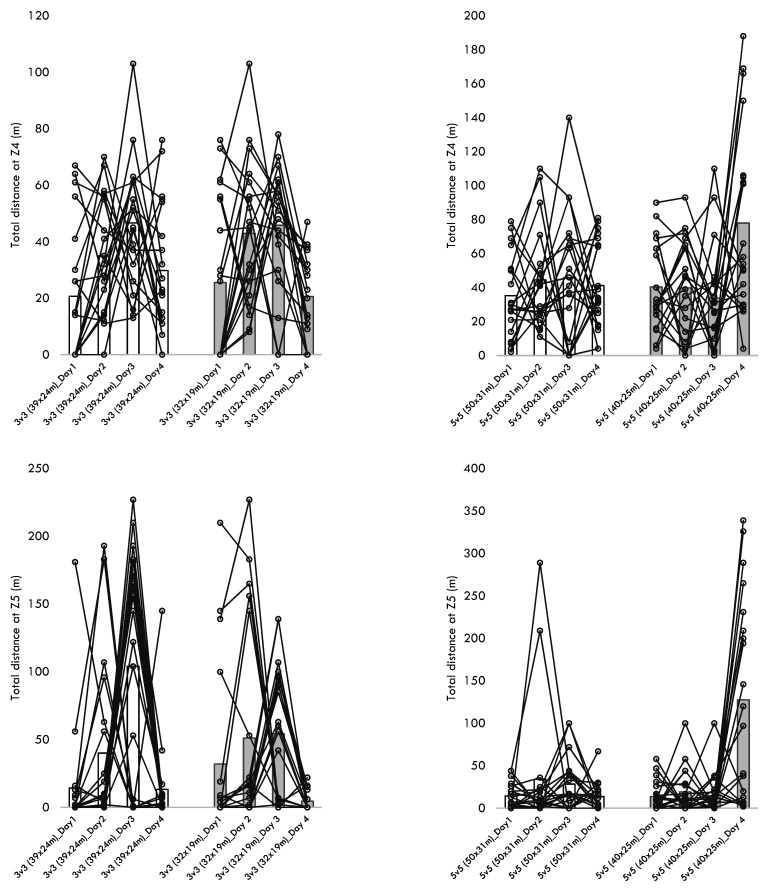
Descriptive statistics of distances covered at Z4 and Z5 over the different formats, pitch dimensions, and days.

**Figure 5 healthcare-10-01412-f005:**
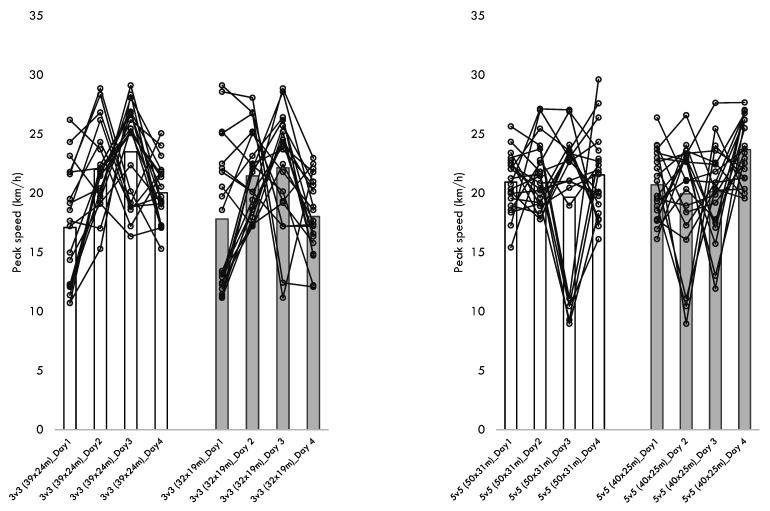
Descriptive statistics of peak speed over the different formats, pitch dimensions, and days.

**Table 1 healthcare-10-01412-t001:** Study design.

		Day 1	Day 2	Day 3	Day 4				Day 1	Day 2	Day 3	Day 4
Match day	Day off	Warm-up FIFA 11+	Warm-up FIFA 11+	Warm-up FIFA 11+	Warm-up FIFA 11+	Day off	Match day	Day off	Warm-up FIFA 11+	Warm-up FIFA 11+	Warm-up FIFA 11+	Warm-up FIFA 11+
		Format: 5v5 Pitch: 50 × 31 m Area per player: 155 m^2^ Length per width ratio: 1.6 Time: 5 min Small goal in the center No offside	Format: 5v5 Pitch: 40 × 25 m Area per player: 100 m^2^ Length per width ratio: 1.6 Time: 5 min Small goal in the center No offside	Format: 5v5 Pitch: 50 × 31 m Area per player: 155 m^2^ Length per width ratio: 1.6 Time: 5 min Small goal in the center No offside	Format: 5v5 Pitch: 40 × 25 m Area per player: 100 m^2^ Length per width ratio: 1.6 Time: 5 min Small goal in the center No offside				Format: 3v3 Pitch: 39 × 24 m Area per player: 156 m^2^ Length per width ratio: 1.6 Time: 3 min Small goal in the center No offside	Format: 3v3 Pitch: 32 × 19 m Area per player: 101 m^2^ Length per width ratio: 1.7 Time: 3 min Small goal in the center No offside	Format: 3v3 Pitch: 39 × 24 m Area per player: 156 m^2^ Length per width ratio: 1.6 Time: 3 min Small goal in the center No offside	Format: 3v3 Pitch: 32 × 19 m Area per player: 101 m^2^ Length per width ratio: 1.7 Time: 3 min Small goal in the center No offside
		Recovery 3 min	Recovery 3 min	Recovery 3 min	Recovery 3 min				Recovery 3 min	Recovery 3 min	Recovery 3 min	Recovery 3 min
		Format: 5v5 Pitch: 40 × 25 m Area per player: 100 m^2^ Length per width ratio: 1.6 Time: 5 min Small goal in the center No offside	Format: 5v5 Pitch: 50 × 31 m Area per player: 155 m^2^ Length per width ratio: 1.6 Time: 5 min Small goal in the center No offside	Format: 5v5 Pitch: 40 × 25 m Area per player: 100 m^2^ Length per width ratio: 1.6 Time: 5 min Small goal in the center No offside	Format: 5v5 Pitch: 50 × 31 m Area per player: 155 m^2^ Length per width ratio: 1.6 Time: 5 min Small goal in the center No offside				Format: 3v3 Pitch: 32 × 19 m Area per player: 101 m^2^ Length per width ratio: 1.7 Time: 3 min Small goal in the center No offside	Format: 3v3 Pitch: 39 × 24 m Area per player: 156 m^2^ Length per width ratio: 1.6 Time: 3 min Small goal in the center No offside	Format: 3v3 Pitch: 32 × 19 m Area per player: 101 m^2^ Length per width ratio: 1.7 Time: 3 min Small goal in the center No offside	Format: 3v3 Pitch: 39 × 24 m Area per player: 156 m^2^ Length per width ratio: 1.6 Time: 3 min Small goal in the center No offside

**Table 2 healthcare-10-01412-t002:** Between-session variability (CV%) in the 3v3 format.

	3v3 (39 × 24 m) Day1 BPV (CV%)	3v3 (39 × 24 m) Day2 BPV (CV%)	3v3 (39 × 24 m) Day3 BPV (CV%)	3v3 (39 × 24 m) Day4 BPV (CV%)	3v3 (32 × 19 m) Day1 BPV (CV%)	3v3 (32 × 19 m) Day 2 BPV (CV%)	3v3 (32 × 19 m) Day 3 BPV (CV%)	3v3 (32 × 19 m) Day 4 BPV (CV%)
HRmean (bpm)	11.0	7.1	5.8	4.6	9.7	6.8	9.2	9.4
HRpeak (bpm)	11.1	3.5	5.1	3.2	3.1	5.1	8.9	3.5
Total distance (m)	33.6	29.0	30.7	20.1	20.8	27.0	14.8	16.4
Distance at Z1 (m)	32.3	50.3	104.4	27.7	41.7	60.0	68.5	17.5
Distance at Z2 (m)	28.3	56.1	41.5	47.7	36.3	48.9	30.2	16.4
Distance at Z3 (m)	94.7	40.0	39.4	26.5	104.4	28.9	60.3	54.1
Distance at Z4 (m)	118.4	58.1	48.9	75.2	113.8	58.6	48.2	72.0
Distance at Z5 (m)	289.8	149.2	82.5	245.0	195.7	148.5	91.1	156.3
Peak speed (km/h)	29.1	16.1	17.5	12.8	34.7	16.2	22.0	17.9

BPV: between-player variability; CV%: coefficient of variation expressed as a percentage.

**Table 3 healthcare-10-01412-t003:** Between-session variability (CV%) in 5v5 format.

	5v5 (50 × 31 m) Day1 BPV (CV%)	5v5 (50 × 31 m) Day2 BPV (CV%)	5v5 (50 × 31 m) Day3 BPV (CV%)	5v5 (50 × 31 m) Day4 BPV (CV%)	5v5 (40 × 25 m) Day1 BPV (CV%)	5v5 (40 × 25 m) Day 2 BPV (CV%)	5v5 (40 × 25 m) Day 3 BPV (CV%)	5v5 (40 × 25 m) Day 4 BPV (CV%)
HRmean (bpm)	10.5	7.4	19.7	8.3	8.5	16.5	13.6	8.5
HRpeak (bpm)	8.6	8.5	16.9	6.6	7.0	15.2	12.4	8.5
Total distance (m)	21.6	14.1	23.6	13.4	18.6	21.0	32.1	17.4
Distance at Z1 (m)	19.8	17.3	24.9	17.6	13.9	16.4	26.7	15.6
Distance at Z2 (m)	40.6	29.0	33.7	25.0	38.5	34.3	40.2	27.5
Distance at Z3 (m)	50.4	40.0	74.8	38.5	42.2	64.2	65.1	51.1
Distance at Z4 (m)	68.0	65.2	84.5	61.7	62.9	73.3	87.5	70.3
Distance at Z5 (m)	103.7	221.1	111.1	116.8	127.1	134.0	156.8	94.0
Peak speed (km/h)	11.9	13.3	30.6	16.6	13.8	24.6	19.3	11.8

BPV: between-player variability; CV%: coefficient of variation expressed as a percentage.

**Table 4 healthcare-10-01412-t004:** Within-player variability (CV%) in both formats and pitch dimensions.

	3v3 (39 × 24 m) All WPV (CV%)	3v3 (32 × 19 m) All WPV (CV%)	5v5 (50 × 31 m) All WPV (CV%)	5v5 (40 × 25 m) All WPV (CV%)	Repeated Measures ANOVA $p|\eta_{p}^{2}$
HRmean (bpm)	7.9 ± 4.0	8.7 ± 4.1	11.9 ± 6.5	11.1 ± 7.1	0.089|0.126
HRpeak (bpm)	6.0 ± 4.3	6.5 ± 4.1	10.3 ± 6.7	10.4 ± 6.3	0.032|0.172
Total distance (m)	34.8 ± 10.5 ^b,c,d^	23.0 ± 7.2 ^a^	18.4 ± 8.4 ^a,d^	26.0 ± 8.2 ^a,c^	<0.001|0.441
Distance at Z1 (m)	60.3 ± 13.8 ^c,d^	53.1 ± 19.1 ^c,d^	19.6 ± 10.1 ^a,b^	20.4 ± 7.7 ^a,b^	<0.001|0.745
Distance at Z2 (m)	52.8 ± 14.0 ^c,d^	46.3 ± 14.4 ^c,d^	32.2 ± 13.5 ^a,b^	32.7 ± 12.7 ^a,b^	<0.001|0.444
Distance at Z3 (m)	51.2 ± 20.5	62.8 ± 20.9	51.1 ± 24.7	52.9 ± 21.9	0.208|0.076
Distance at Z4 (m)	77.5 ± 31.7	83.0 ± 29.1	68.3 ± 23.8	79.4 ± 33.9	0.457|0.044
Distance at Z5 (m)	170.5 ± 27.6 ^c,d^	158.6 ± 32.3 ^c^	113.7 ± 38.5 ^a,b^	139.7 ± 33.9 ^a^	<0.001|0.394
Peak speed (km/h)	22.1 ± 9.5	25.0 ± 9.9	17.0 ± 10.5	18.6 ± 8.8	0.052|0.146

WPV: within-player variability (average of within-players repeated measures across the days); CV%: coefficient of variation expressed as percentage; ^a^: significantly different from 3v3 (39 × 24 m) at *p* < 0.05; ^b^: significantly different from 3v3 (32 × 19 m) at *p* < 0.05; ^c^: significantly different from 5v5 (50 × 31m) at *p* < 0.05; ^d^: significantly different from 5v5 (40 × 25 m) at *p* < 0.05.

**Table 5 healthcare-10-01412-t005:** Descriptive statistics (mean and standard deviation) of heart rate and locomotor demands between sessions in regard to the 3v3 format.

	**3v3 (39 × 24 m)** **Day1**	**3v3 (39 × 24 m)** **Day2**	**3v3 (39 × 24 m)** **Day3**	**3v3 (39 × 24 m)** **Day4**	**Repeated Measures ANOVA** $p|\eta_{p}^{2}$
HRmean (bpm)	157.0 ± 17.3 ^c,d^	170.4 ± 12.1	172.8 ± 10.1 ^a^	174.8 ± 8.0 ^a^	0.001|0.303
Hrpeak (bpm)	176.2 ± 19.6	187.0 ± 6.5	184.2 ± 9.4	189.4 ± 6.1	0.023|0.189
Total distance (m)	324.3 ± 108.9 ^c^	384.6 ± 111.7	489.5 ± 150.4 ^a,d^	343.9 ± 69.1 ^c^	0.005|0.281
Distance at Z1 (m)	79.0 ± 25.5 ^c,d^	78.2 ± 39.3 ^d^	41.1 ± 42.9 ^a,d^	115.2 ± 31.8 ^a,b,c^	0.001|0.422
Distance at Z2 (m)	153.6 ± 43.5 ^d^	165.5 ± 92.8	215.9 ± 89.5 ^d^	112.4 ± 53.6 ^a,c^	0.007|0.255
Distance at Z3 (m)	50.5 ± 47.8	61.9 ± 24.7	78.1 ± 30.8	65.9 ± 17.5	0.092|0.116
Distance at Z4 (m)	20.7 ± 24.5 ^c^	35.0 ± 20.3	46.7 ± 22.8 ^a^	29.8 ± 22.4	0.006|0.196
Distance at Z5 (m)	14.3 ± 41.3 ^c^	40.1 ± 59.8	104.1 ± 85.8 ^a,d^	13.3 ± 32.5 ^c^	0.002|0.324
Peak speed (km/h)	17.1 ± 5.0 ^b,c^	22.0 ± 3.6 ^a^	23.5 ± 4.^1 a,d^	20.0 ± 2.6 ^c^	<0.001|0.330
	**3v3 (32 × 19 m)** **Day1**	**3v3 (32 × 19 m)** **Day 2**	**3v3 (32 × 19 m)** **Day 3**	**3v3 (32 × 19 m)** **Day 4**	**Repeated Measures ANOVA** $p|\eta_{p}^{2}$
Hrmean (bpm)	162.6 ± 15.8	169.0 ± 11.4	173.1 ± 15.9	168.2 ± 15.9	0.213|0.075
Hrpeak (bpm)	177.9 ± 17.6	186.7 ± 5.8	187.3 ± 9.6	182.7 ± 16.3	0.138|0.100
Total distance (m)	409.9 ± 85.4 ^d^	435.8 ± 117.7 ^d^	454.8 ± 67.4 ^d^	331.3 ± 54.3 ^a,b,c^	0.003|0.301
Distance at Z1 (m)	76.5 ± 31.9 ^d^	76.8 ± 46.1	53.4 ± 36.6 ^d^	103.1 ± 18.1 ^a,c^	0.007|0.259
Distance at Z2 (m)	233.6 ± 84.9 ^d^	172.3 ± 84.1	245.7 ± 74.2 ^d^	123.0 ± 20.2 ^a,c^	0.001|0.404
Distance at Z3 (m)	37.5 ± 39.2 ^b^	84.9 ± 24.5 ^a,c^	53.9 ± 32.5 ^b^	71.2 ± 38.5	0.001|0.289
Distance at Z4 (m)	25.6 ± 29.1	43.0 ± 25.2 ^d^	45.9 ± 22.1 ^d^	20.6 ± 14.8 ^b,c^	0.009|0.220
Distance at Z5 (m)	32.0 ± 62.6	51.2 ± 76.0	54.3 ± 49.5 ^d^	4.6 ± 7.1 ^c^	0.066|0.151
Peak speed (km/h)	17.8 ± 6.2 b	21.5 ± 3.5 ^a,d^	22.1 ± 4.9 ^d^	18.0 ± 3.2 ^b,c^	0.037|0.180

^a^: significantly different from 3v3 (39 × 24 m) at *p* < 0.05; ^b^: significantly different from 3v3 (32 × 19 m) at *p* < 0.05; ^c^: significantly different from 5v5 (50 × 31 m) at *p* < 0.05; ^d^: significantly different from 5v5 (40 × 25 m) at *p* < 0.05.

**Table 6 healthcare-10-01412-t006:** Descriptive statistics (mean and standard deviation) of heart rate and locomotor demands between sessions in regard to the 5v5 format.

	**5v5 (50 × 31 m)** **Day1**	**5v5 (50 × 31 m)** **Day2**	**5v5 (50 × 31 m)** **Day3**	**5v5 (50 × 31 m)** **Day4**	**Repeated Measures ANOVA** $p|\eta_{p}^{2}$
HRmean (bpm)	164.6 ± 17.3	165.4 ± 12.2	154.9 ± 30.5	160.8 ± 13.3	0.357|0.053
Hrpeak (bpm)	182.3 ± 15.7	190.0 ± 16.1	170.9 ± 28.9	183.9 ± 12.2	0.062|0.136
Total distance (m)	486.2 ± 104.9	555.8 ± 78.3	529.8 ± 124.8	531.8 ± 71.0	0.197|0.078
Distance at Z1 (m)	199.8 ± 39.5	212.8 ± 36.8	207.5 ± 51.6	181.9 ± 32.1	0.128|0.100
Distance at Z2 (m)	125.6 ± 51.0	148.1 ± 43.0	148.5 ± 50.0	156.0 ± 38.9	0.187|0.080
Distance at Z3 (m)	84.6 ± 42.6	96.5 ± 38.6	86.5 ± 64.7	112.6 ± 43.4	0.263|0.067
Distance at Z4 (m)	35.3 ± 24.0	44.5 ± 29.0	45.7 ± 38.6	41.3 ± 25.5	0.703|0.024
Distance at Z5 (m)	14.4 ± 14.9	34.1 ± 75.3	28.4 ± 31.6	13.8 ± 16.1	0.323|0.056
Peak speed (km/h)	20.9 ± 2.5	21.5 ± 2.9	19.7 ± 6.0	21.5 ± 3.6	0.389|0.049
	**5v5 (40 × 25 m)** **Day1**	**5 v5 (40 × 25 m)** **Day 2**	**5v5 (40 × 25 m)** **Day 3**	**5v5 (40 × 25 m)** **Day 4**	**Repeated Measures ANOVA** $p|\eta_{p}^{2}$
Hrmean (bpm)	168.6 ± 14.3	160.8 ± 26.6	163.4 ± 22.3	166.4 ± 14.1	0.668|0.027
Hrpeak (bpm)	183.6 ± 12.9	179.7 ± 27.2	181.7 ± 22.4	190.1 ± 16.2	0.464|0.044
Total distance (m)	522.0 ± 97.0	522.6 ± 109.7	413.9 ± 132.9	596.8 ± 103.8	<0.001|0.291
Distance at Z1 (m)	207.1 ± 28.8	227.2 ± 37.4	174.4 ± 46.5	176.9 ± 27.6	<0.001|0.377
Distance at Z2 (m)	146.8 ± 56.5	137.5 ± 47.2	108.1 ± 43.5	134.0 ± 36.8	0.054|0.124
Distance at Z3 (m)	94.1 ± 39.7 ^d^	82.6 ± 53.1	65.1 ± 42.4	64.4 ± 32.9 ^a^	0.075|0.113
Distance at Z4 (m)	40.4 ± 25.4	39.8 ± 29.2	33.9 ± 29.6	78.0 ± 54.8	0.008|0.225
Distance at Z5 (m)	13.8 ± 17.5 ^d^	19.2 ± 25.7 ^d^	14.8 ± 23.2 ^d^	127.6 ± 119.9 ^a,b,c^	<0.001|0.438
Peak speed (km/h)	20.7 ± 2.9 ^d^	20.0 ± 4.9 ^d^	20.1 ± 3.9 ^d^	23.7 ± 2.^8 a,b,c^	0.012|0.174

^a^: significantly different from 3v3 (39 × 24 m) at *p* < 0.05; ^b^: significantly different from 3v3 (32 × 19 m) at *p* < 0.05; ^c^: significantly different from 5v5 (50 × 31 m) at *p* < 0.05; ^d^: significantly different from 5v5 (40 × 25 m) at *p* < 0.05.

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
