# Peer review of "Analyzing the within and between Players Variability of Heart Rate and Locomotor Responses in Small-Sided Soccer Games Performed Repeatedly over a Week"

_healthcare, 2022, doi:10.3390/healthcare10081412_

Round 1

Reviewer 1 Report

The objective of this study was to analyze the heart rate and locomotor variability of young male soccer players in 3v3 and 5v5 small-sided games both across sessions and between players. These findings will be helpful to plan the training prescription for coaches.

However, the authors should address some concerns before we can accept the article for publication.

1. Please use the full name when ‘SSGs’ appears for the first time in the “Abstract” section.

2. As mentioned in the introduction, the time of the day also has a certain impact on the performance of young players, is the experiment in the paper conducted at different times of the day?

3. Is the ‘average heart rate (HRav)’ and ‘HRmean’ mentioned in the paper the same indicator? If it is the same, please use one name to refer to it. Is ‘peak heart rate (HRpeak)’ and ‘HRmax’ the same indicator? If it is the same, please use one name to refer to it.

4. These publications may be helpful to your future research

The Reproducibility of Physiological Responses and Performance Profiles of Youth Soccer Players in Small-Sided Games. International Journal of Sports Physiology and Performance, 2008, 3, 393-396

Variability of Technical Actions During Small‐Sided Games in Young Soccer Players.Journal of Human Kinetics volume 69/2019, 201‐212

Biomechanical performance design of joint prosthesis for medical rehabilitation via generative structure optimization. Computer Methods in Biomechanics and Biomedical Engineering, 2020, 23(15): 1163-1179.

Biomechanical strengthening design for limb articulation based on reconstructed skeleton kinesthetics. Journal of Medical and Biological Engineering, 2021.

Author Response

Response to question 1: Dear reviewer, thank you. We have changed accordingly.

Response to question 2: Dear reviewer, thank you. The information about the exact time can be observed in the section 2.1. Experimental approach to the problem

Response to question 3: Dear reviewer, thank you. We have standardized all manuscript to HRmean and HRpeak

Question 4: Dear reviewer, thank you. We have added two of the suggested references aiming to support the introduction.

Reviewer 2 Report

An innovative element of this paper is to show the variability of the heart rate, length and speed of the distances covered during small games and the differences resulting in these ranges between the 3v3 and 5v5 games. The study showed that heart rate is a relatively stable indicator that may be of importance in training management. This index can be used to reliably assess your aerobic performance. However, physical performance in soccer depends on both aerobic capacity and anaerobic capacity. Therefore, at least in the introduction, it should be noted that to illustrate the full physical performance, one should also track the indicators of anaerobic power and their variability, as it was done in relation to the  heart rate or the length of the distance run. It seems that the most reliable indicator in this respect would be the measurement of blood lactate concentration variability. Such an examination is more difficult to perform, but it is possible.

Author Response

An innovative element of this paper is to show the variability of the heart rate, length and speed of the distances covered during small games and the differences resulting in these ranges between the 3v3 and 5v5 games. The study showed that heart rate is a relatively stable indicator that may be of importance in training management. This index can be used to reliably assess your aerobic performance. However, physical performance in soccer depends on both aerobic capacity and anaerobic capacity.

AUTHORS: DEAR REVIEWER, WE WOULD LIKE TO THANK YOU FOR THE COMMENTS AND SUGGESTIONS PROVIDED TO US. WE BELIEVE THAT YOUR FEEDBACK HELPED US TO IMPROVE THE ARTICLE.

Therefore, at least in the introduction, it should be noted that to illustrate the full physical performance, one should also track the indicators of anaerobic power and their variability, as it was done in relation to the  heart rate or the length of the distance run. It seems that the most reliable indicator in this respect would be the measurement of blood lactate concentration variability. Such an examination is more difficult to perform, but it is possible.

AUTHORS: DEAR REVIEWER, THANK YOU SO MUCH. WE DID NOT ANALYZE THAT. BUT WE DO AGREE WITH YOU. THUS, WE HAVE ADDED TO THE DISCUSSION, NAMELY IN STUDY LIMITATIONS AND FUTURE RESEARCH.

Reviewer 3 Report

The authors present the findings of a small repeated-measures study design of 20 junior soccer players convenience sampled from one team completing small format 3x3 and 5x5 drills conducted in the early phases of the season. The findings indicate that there is less heart rate variability between players in the 3x3 format but greater locomotor variability, and conversely greater HR variability in the 5x5 format with less locomotor variability. The authors conclude that 5x5 may be a better choice for stabilising locomotor demands, while 3x3 for HR demands. 

1. I have significant concerns about the English and grammar within this article. There are frequent grammatical and phrasing errors throughout this article in all sections that require substantial editing. I suggest that the authors seek and utilise an English editing service to improve this manuscript as it is not at a sufficient standard for publication.

2. In my opinion, the authors do not adequately justify their rationale for conducting this study in a small sample of junior soccer players in the introduction. Fundamentally, It is unclear why knowledge of variability in HR and locomotor demands between players in the 3x3 and 5x5 format would be of interest to coaches and training staff - as load and injury management takes place at an individual level - i.e. The volume of training for each individual should be managed within the team environment using the parameters collected for each individual. 

3. There also does not appear to be enough control around the baseline parameters of individuals enrolled within the study. For example, in the methods, the prior training volume of participants is estimated, not measured and is a potential confounder to the results observed. Additionally, there is no baseline aerobic capacity testing data provided nor sprint data to provide context for the HR values observed and sprint capacity. This is fundamental to determining the intensity of training - i.e. a HR of 170bpm will have varying relative intensity depending on the aerobic capacity of each individual. Similarly, assigning Z5 as the 'sprint' zone may not be appropriate for each individual as varying biomechanics and physiological factors may ultimately change what a 'sprint' is for each player. 

4. The conclusion does not flow well and the limitations are not discussed sufficiently. There needs to be more discussion around the limitations and the conclusions made by the authors needs more caution as a result of these limitations. Additionally, there is no discussion either in the introduction or conclusion of what actually are the ideal workloads and intensities for sub-elite and elite level soccer/football players engaged in training. 

Thank you. 

Author Response

The authors present the findings of a small repeated-measures study design of 20 junior soccer players convenience sampled from one team completing small format 3x3 and 5x5 drills conducted in the early phases of the season. The findings indicate that there is less heart rate variability between players in the 3x3 format but greater locomotor variability, and conversely greater HR variability in the 5x5 format with less locomotor variability. The authors conclude that 5x5 may be a better choice for stabilising locomotor demands, while 3x3 for HR demands. 

AUTHORS: DEAR REVIEWER, WE WOULD LIKE TO THANK YOU FOR THE COMMENTS AND SUGGESTIONS PROVIDED TO US. WE BELIEVE THAT YOUR FEEDBACK HELPED US TO IMPROVE THE ARTICLE.

  1. I have significant concerns about the English and grammar within this article. There are frequent grammatical and phrasing errors throughout this article in all sections that require substantial editing. I suggest that the authors seek and utilise an English editing service to improve this manuscript as it is not at a sufficient standard for publication.

AUTHORS: DEAR REVIEWER, THANK YOU. WE ALSO AGREE WITH YOU. HOWEVER, IF ACCEPTED, THE ARTICLE IS FULLY REVISED BY MDPI PROOFREADING SERVICE, WHICH IS A ENSURANCE FOR US AND GUARANTEES THAT THE ARTICLE WILL BE READY FOR THE COMMUNITY.

  1. In my opinion, the authors do not adequately justify their rationale for conducting this study in a small sample of junior soccer players in the introduction. Fundamentally, It is unclear why knowledge of variability in HR and locomotor demands between players in the 3x3 and 5x5 format would be of interest to coaches and training staff - as load and injury management takes place at an individual level - i.e. The volume of training for each individual should be managed within the team environment using the parameters collected for each individual. 

AUTHORS: DEAR REVIEWER, THANK YOU. WE DO AGREE. WE HAVE ADDED A STATEMENT OF CONTRIBUTION IN THE LAST PARAGRAPH OF THE INTRODUCTION.

  1. There also does not appear to be enough control around the baseline parameters of individuals enrolled within the study. For example, in the methods, the prior training volume of participants is estimated, not measured and is a potential confounder to the results observed. Additionally, there is no baseline aerobic capacity testing data provided nor sprint data to provide context for the HR values observed and sprint capacity. This is fundamental to determining the intensity of training - i.e. a HR of 170bpm will have varying relative intensity depending on the aerobic capacity of each individual. Similarly, assigning Z5 as the 'sprint' zone may not be appropriate for each individual as varying biomechanics and physiological factors may ultimately change what a 'sprint' is for each player. 

AUTHORS: DEAR REVIEWER, THANK YOU. IN FACT, WE PERFORMED A INITIAL ASSESSMENT IN THE WEEK PRIOR TO DATA COLLECTION. THE 30-15IFT AND 30-M SPRINT TEST WERE ASSESSED. WE HAVE ADDED THIS INFORMATION AND THE PRESENTATION OF RESULTS IN THE SECTIONS 2.3 AND 3. MOREOVER, WE HAVE DISCUSSED THE POSSIBLE LIMITATIONS YOU EMPHASIZED IN THE BOTTOM OF THE DISCUSSION (STUDY LIMITATIONS).

  1. The conclusion does not flow well and the limitations are not discussed sufficiently. There needs to be more discussion around the limitations and the conclusions made by the authors needs more caution as a result of these limitations. Additionally, there is no discussion either in the introduction or conclusion of what actually are the ideal workloads and intensities for sub-elite and elite level soccer/football players engaged in training. 

AUTHORS: DEAR REVIEWER, THANK YOU. WE HAVE EXTENDED THE STUDY LIMITATIONS BASED ON YOUR SUGGESTIONS. ALSO, CHANGES WERE PERFORMED IN THE CONCLUSION. WE CANNOT DISCUSS DOSE OR TRAINING VOLUME SINCE WE DO HAVE DATA FOR THIS AND THE RESEARCH WAS NOT PREPARED FOR THAT.

Thank you.

Reviewer 4 Report

The purpose of this study was to analyze the between-session and within-player variability of heart rate and locomotor responses of youth male soccer players in 3v3 and 5v5 small- sided games format. Thank you for the opportunity to review you manuscript. I have a few comments: 

Abstract: Extend the introduction. In addition, the abstract should be divided into the following sketches: Backgroud, Methods, Results, Conclusion 

The introduction is well-written and provides a rationale for the issues raised 

Material and method: How many subjects were recruited? How many were rejected?  Inclusion and exclusion criteria can be given in a table. Additionally, how was the sample size calculated? Why two hundred participants? . It would be worth mentioning which specific breast strap model was used. 

Statistical methods: The analysis seems to have been carried out correctly. Was the conformity of the distribution with the normal distribution checked? it is worth mentioning. 

Results: Reference to table number 3 and figures is missing, this should be included in the text 

Limitations of the research should be written in italics like a subsection 

Does your research have any practical application? It is worth developing an application 

delete section number 6 

Author Response

The purpose of this study was to analyze the between-session and within-player variability of heart rate and locomotor responses of youth male soccer players in 3v3 and 5v5 small- sided games format. Thank you for the opportunity to review you manuscript.

AUTHORS: DEAR REVIEWER, WE WOULD LIKE TO THANK YOU FOR THE COMMENTS AND SUGGESTIONS PROVIDED TO US. WE BELIEVE THAT YOUR FEEDBACK HELPED US TO IMPROVE THE ARTICLE.

I have a few comments: 

Abstract: Extend the introduction. In addition, the abstract should be divided into the following sketches: Backgroud, Methods, Results, Conclusion 

AUTHORS: DEAR REVIEWER, THANK YOU. WE HAVE INCLUDED A BACKGROUND SECTION IN THE ABSTRACT. 

The introduction is well-written and provides a rationale for the issues raised 

AUTHORS: DEAR REVIEWER, THANK YOU SO MUCH FOR THE KIND COMMENT.

Material and method: How many subjects were recruited? How many were rejected?  Inclusion and exclusion criteria can be given in a table. Additionally, how was the sample size calculated? Why two hundred participants? . It would be worth mentioning which specific breast strap model was used. 

AUTHORS: DEAR REVIEWER, THANK YOU. WE HAVE ADDED IN SECTION 2.2., THE SAMPLE SIZE CALCULATION AND THE DETAILS OF INITIAL RECRUITMENT AND EXCLUSION. 

Statistical methods: The analysis seems to have been carried out correctly. Was the conformity of the distribution with the normal distribution checked? it is worth mentioning. 

AUTHORS: DEAR REVIEWER, THANK YOU. WE HAVE ADDED THE DETAIL IN SECTION 2.5. 

Results: Reference to table number 3 and figures is missing, this should be included in the text 

AUTHORS: DEAR REVIEWER, THANK YOU. REFERENCE TO TABLE 3 IS PRESENT IN THE FIRST PARAGRAPH OF THE RESULTS. THE REFERENCE TO FIGURES IS PRESENT IN PARAGRAPH 3.

Limitations of the research should be written in italics like a subsection 

AUTHORS: DEAR REVIEWER, THANK YOU. WE HAVE ADDED AS A SUB-SECTION NOW. 

Does your research have any practical application? It is worth developing an application 

AUTHORS: DEAR REVIEWER, THANK YOU. WE HAVE ADDED A NEW SECTION (4.2., PRACTICAL APPLICATIONS). 

delete section number 6 

AUTHORS: DEAR REVIEWER, THANK YOU. THE SECTION WAS REMOVED.

Round 2

Reviewer 3 Report

The authors have made significant improvements to the manuscript, and while I still believe the relevance and external validity of the findings from this small, sub-elite, junior sample is low - the robustness of the methodology has improved dramatically.

The authors still do not however adequately describe their limitations in the context of the small sample and must be more precise with the risk of bias that may result and the implications for interpretation of these findings as a result. The authors should be less finite in their conclusions due to the low power and external validity of their study, and instead provide more details to their current analysis on how future studies could be conducted to test and evaluate these findings with greater rigours and external validity, while acknowledging the contribution this study is likely to make. Furthermore, they should address what questions remain and have not yet been answered in relation to their study. 

Currently, the quality of English writing throughout the manuscript must be improved prior to submission, regardless of whether the article is due to undergo english editing. This is essential for understanding the key points made by the authors. 

Author Response

The authors have made significant improvements to the manuscript, and while I still believe the relevance and external validity of the findings from this small, sub-elite, junior sample is low - the robustness of the methodology has improved dramatically.

AUTHORS: DEAR REVIEWER, THANK YOU SO MUCH FOR PROVIDING US A POSITIVE FEEDBACK AND HELPING US WITH CONSTRUCTIVE COMMENTS WHICH HELPED US TO IMPROVE THE DOCUMENT.

The authors still do not however adequately describe their limitations in the context of the small sample and must be more precise with the risk of bias that may result and the implications for interpretation of these findings as a result.

AUTHORS: DEAR REVIEWER, THANK YOU. WE HAVE ADDED THIS IN THE STUDY LIMITATIONS “The small sample included is one of the limitations that should be considered as a potential bias. Another limitation is convenience sampling. All the players were part of the same team, which should be faced as a bias for generalization of the evidence. Thus, any conclusion should not be observed as an irrefutable finding and requires more observations to be confirmed.”.

The authors should be less finite in their conclusions due to the low power and external validity of their study, and instead provide more details to their current analysis on how future studies could be conducted to test and evaluate these findings with greater rigours and external validity, while acknowledging the contribution this study is likely to make. Furthermore, they should address what questions remain and have not yet been answered in relation to their study. 

AUTHORS: DEAR REVIEWER, THANK YOU. WE HAVE ADDED THIS IN THE CONCLUSIONS “The results come from a small sample and convenience sampling. Thus, these results are context-dependent which should be faced as a limitation for the generalization of the findings. Future studies should be conducted to confirm or not the evidence. The next studies must include more players to increase the sample size. Additionally, different teams and populations must be included for a more robust analysis. Finally, consideration for moderators and mediators as training status, skill level, and moment of the season must be also included.